SOFTWARE

# ShapeSpaceExplorer: Analysis of morphological transitions in migrating cells using similarity-based shape space mapping

Samuel D. R. Jefferyes[1,2], Roswitha Gostner[1¤a], Laura Cooper[1],
Mohammed M. Abdelsamea[1¤b], Elly Straube[1¤c], Nasir Rajpoot[3],
David B. A. Epstein[4], Anne Straube[1]*

1 Centre for Mechanochemical Cell Biology and Biomedical Sciences, Warwick Medical School, University of Warwick, Coventry, United Kingdom, 2 Systems Biology Doctoral Training Centre, University of Warwick, Coventry, United Kingdom, 3 Department of Computer Science, University of Warwick, Coventry, United Kingdom, 4 Mathematics Institute, University of Warwick, Coventry, United Kingdom

¤a Current address: Carl Zeiss Digital Innovation GmbH, Munich, Germany
¤b Current address: Department of Computer Science, Faculty of Environment, Science and Economy, University of Exeter, Streatham Campus, Exeter, United Kingdom
¤c Current address: Institute of Natural Materials Technology, Dresden University of Technology, Dresden, Germany
* anne@mechanochemistry.org

## Abstract

Here we describe the development of ShapeSpaceExplorer, an interactive software for the extraction, visualisation and analysis of complex 2D shape series. We also demonstrate its application to the analysis of cell morphology changes during cell migration. Cell migration is essential for many physiological and pathological processes. Intracellular force generation and transmission of these forces to extracellular structures or neighbouring cells drives cell migration. The emergent property of the processes driving cell migration is a change in cell shape. We describe a machine learning approach to understand the relationship of cell shape dynamics and cell migration behaviour. Our algorithm analyses cell shape from time-lapse images and learns the intrinsic low-dimensional structure of cell shape space. We use the resultant shape space map to visualise differences in cell shape distribution following perturbation experiments and to analyse the quantitative relationships between shape and migration behaviour. The core of our algorithm is a new, rapid, and landmark-free shape difference measure that allows unbiased analysis of the widely varying morphologies exhibited by migrating mesenchymal cells. We used our method to predict cell turning from dynamic cell shape information. ShapeSpaceExplorer can be applied widely to visualise and analyse cell morphology changes during development, the cell cycle and stress response, but also to the outlines of clusters, tissues and inanimate objects.

**Data availability statement:** The code and documentation is available at https://github.com/cmcb-warwick/ShapeSpaceExplorer and in the Supporting information files. The raw image data used for this study are available at DOI 10.5281/zenodo.14998398 and the processed data are available at DOI 10.5281/zenodo.14999395.

**Funding:** This work was supported by a PhD studentship in the Warwick Systems Biology Doctoral Training Centre supported by BBSRC and EPSRC (to SDRJ), a postdoctoral fellowship from the Institute of Advanced Study, University of Warwick (to RG), a Lister Institute of Preventive Medicine Research Prize (to AS, funding the salary of MMA and a Lister summer studentship of ES), and a Wellcome Investigator Award in Science (224563/Z/21/Z to AS). Funder details: Biotechnology and Biological Sciences Research Council (BBSRC): https://www.ukri.org/councils/bbsrc/. Engineering and Physical Sciences Research Council (EPSRC): https://www.ukri.org/councils/epsrc/. Lister Institute of Preventive Medicine: https://lister-institute.org.uk/. Wellcome Trust: https://wellcome.org/. Institute of Advanced Study, University of Warwick: https://warwick.ac.uk/fac/cross_fac/ias/. The funders had no role in study design, data collection and analysis, decision to publish, or preparation of the manuscript.

**Competing interests:** The authors have declared that no competing interests exist.

## Introduction

Conventional analysis of cell morphology involves the measurement of shape features such as length, area, circularity and symmetry. Such an analysis is particularly efficient to determine how a specific property of cell shape relates to a specific cellular function. For example, cell length has been found to be an informative feature in the analysis of the *S. pombe* life cycle [1]. Simple shape features can also be used in dynamic models. We have previously described that although the maximum length of cell tails in human epithelial cells remains unchanged, the lifetime of individual tails decreases dramatically when trailing cell adhesion is reduced [2]. To quantify such behaviour usually requires prior knowledge of those features that best describe the different shapes in the dataset. Alternatively, a large number of shape features can be computed and principal component analysis (PCA) or neural network analysis performed to reduce the dimensionality of the dataset [3,4]. However, there are limitations to the suitability of PCA if the investigated data structure is not linear. In addition, feature-based measurements have other problems relating to the fact that objects can have similar features whilst being visually different. Other methods that have been implemented are selecting modes of shape variability using principal component analysis [5], spherical harmonics [6], signed morphomigrational angle [7], *InShaDe* based on discrete curvature along closed, resampled contours [8] and lobe contribution elliptical Fourier analysis (LOCO-EFA) [9].

Mesenchymal cells present a challenging problem in computer vision as their widely varying morphologies are difficult to model, and clearly their shape dynamics are integral for their function and migratory behaviour. In this paper, we develop a quantitative representation of the shape distribution of migrating cells independently of shape descriptors by computing a similarity score for every pair of shapes in the dataset and employ the Diffusion Maps technique [10] to generate a geometric description of the cell shape space. The trajectories of individual cells on the resulting shape space map are then explored to quantify cell shape changes and correlate cell shape dynamics with migratory behaviour. We find that analysis of shape dynamics alone permits the prediction of directional changes during cell migration.

This method is implemented in MATLAB with graphical user interfaces to allow users to intuitively analyse and explore their data. ShapeSpaceExplorer includes tools that allow users to visualise, classify and interact with the results.

## Design and implementation

### Shape space creation

The classic approach to surveying and mapping of land is to measure the distance between objects locally and plot these to scale. Similarly to this, we construct a map of shape space, i.e. a geometric representation of all cell shapes in a dataset, by determining the difference between the shapes in the dataset and learning the local data structure. In the resulting shape space map, points that are close represent shapes that are similar, and points that are far apart represent dissimilar shapes. To implement this for the shapes of migrating cells, we first segment the cell contours from images. Secondly, the cell contours are converted into a descriptor

representation (i.e. the system can be modelled algebraically). Thirdly, we calculate the shape difference between every pair of shapes in the dataset and assimilate these into a large shape similarity matrix. Finally, from this matrix a low-dimensional, quantitative representation of the dataset is created - our shape space map.

**Contour extraction.** The outline of cells from image sequences are extracted using a mean shift algorithm [11]. To overcome any limitations of the mean shift algorithm as a pixel-based segmentation approach, an interactive interface is provided as part of the ShapeSpaceExplorer software to enable manual editing (Fig 1a). Cell segmentations can also be imported, if they have been created using another method or software.

**Cell tracking.** To separate analysis of the cell migration from analysis of the shape of the cells, the software defines a cell's track to be the path of the centroid of the cell over the course of the image sequences. The centroid is computed as the mean of all pixel positions contained within the cell segmentation. As a preprocessing step, cell tracks with missing frames (either because the cell was touching another cell or the boundary could not be segmented) are automatically split and assigned distinct cell IDs for each continuous track segment. The user can filter the tracks for inclusion in the analysis by a minimum number of frames. We used tracks with at least 5 frames for the datasets analysed here. However, outlines from still images can be included in the morphology analysis.

**Shape similarity measurement with Best Alignment Metric (BAM).** In order to provide a robust solution, the software does not rely on any *a priori* information. ShapeSpaceExplorer first approximates each closed curve by a sequence of N complex numbers whose mean lies at zero. The software then utilises a shape similarity measure to compute pairwise alignments of shapes, which we call Best Alignment Metric (BAM). BAM is a novel shape distance measure specifically designed for the comparison of independent simple closed planar curves. This will find the pairwise distance between corresponding pairs of points on the curves after the curves have been mutually aligned, reparameterised and interpolated so as to best emphasize the similarities between the curves (Fig 1b). The formulation employs the Fourier transforms of the curves and circular convolution to efficiently compute the overall optimum solution.

The proposed BAM for discrete curves $v, v = \{v_j, v_j \in \mathbb{C} : j = 0, \dots, N-1\}$, where N is the number of evenly distributed points used to represent each curve, is defined as

$$d^{BAM}([v],[v])^2 = \frac{1}{N} \min_{(r,\theta)} \sum_{j=0}^{N-1} \left| v_j - e^{i\theta} v_{j+r} \right|^2 \tag{1}$$

where $r$ is the cyclic shift of the starting point around the curve, $\theta$ is the rotation angle of the plane, the index $j+r$ is taken modulo $N$, and the minimum is taken over $\{0, \dots, N-1\} \times [0, 2\pi)$. The main advantage of the proposed alignment metric is that it can be rapidly implemented by being reformulated to the following expression,

$$N d^{BAM}([v],[v])^2 = \sum_{j=0}^{N-1} |v_j|^2 + \sum_{j=0}^{N-1} |v_j|^2 - 2 \max_r \sum_{j=0}^{N-1} |v_j \overline{v_{j+r}}|^2. \tag{2}$$

The reasons that this admits a rapid implementation are threefold. Firstly, many of the terms depend only on one curve and so can be computed only once per curve, not per pair of curves. Secondly, $\theta$ is removed, as this formulation explicitly computes the appropriate quantity over all rotations. Thirdly, the last term in the expression can be rapidly computed through use of circular convolution. The fast implementation of BAM allows processing of large datasets.

Algorithm 1 presents the pseudocode for measuring BAM over a large dataset of curves. It highlights the fact that many of the terms can be calculated once for each curve, rather than each pair of curves, which saves a large calculation cost.

While retaining all spatiotemporal information for downstream analysis, to determine the distance between each pair of shapes, BAM only considers the cell outlines normalised to closed curves with the same number of equal spaced nodes. During development of the software, it was found that 512 nodes are sufficient to accurately describe the cell

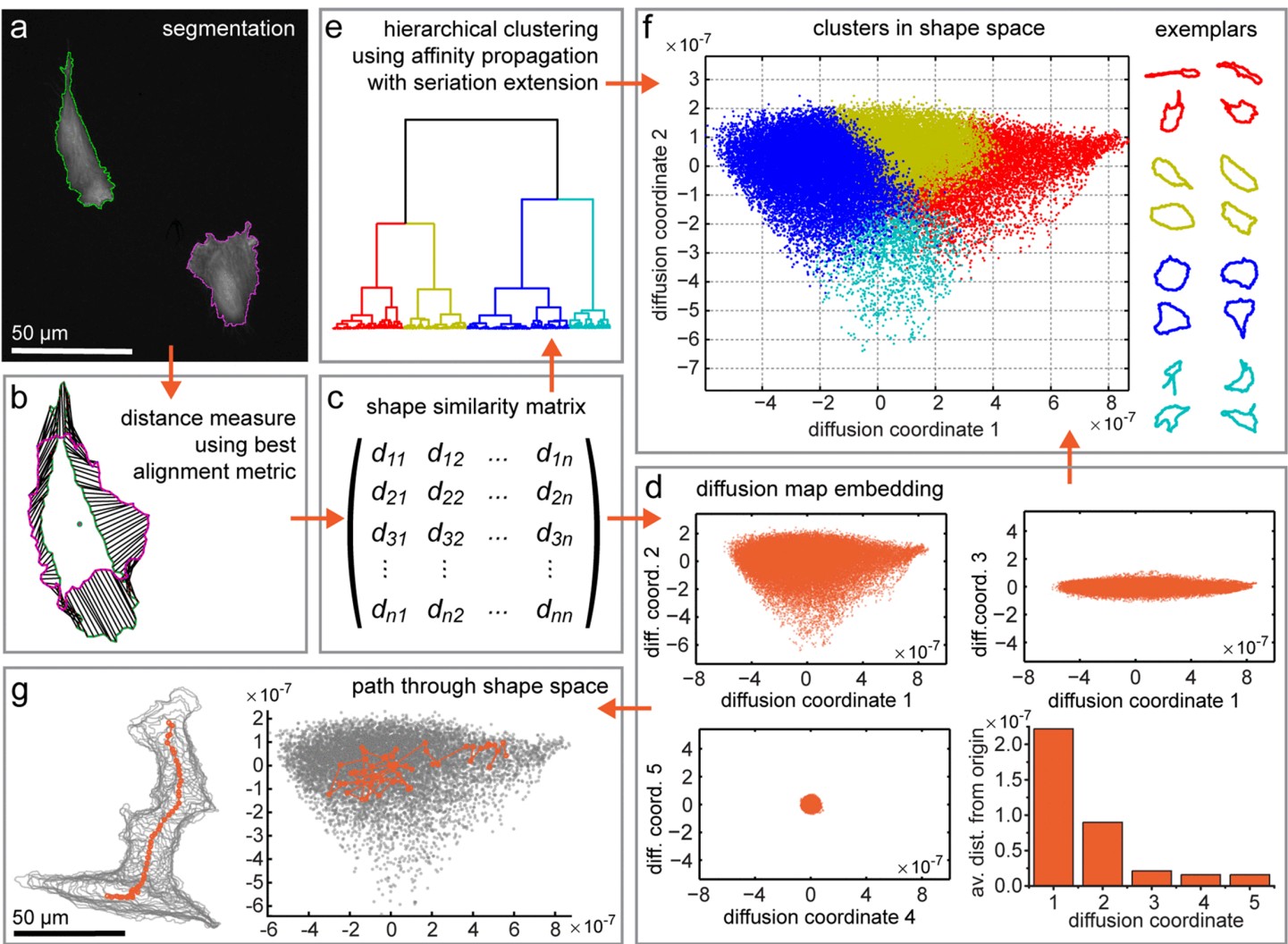

**Fig 1**. **ShapeSpaceExplorer workflow.** Visual representation of the workflow by ShapeSpaceExplorer software. a: Images of RPE1 cells stably expressing GFP-LifeAct are segmented using a mean shift algorithm and manual post-processing. b: Each closed curve representing the cell shape is approximated by 512 nodes. The differences between the curves are measured using the best alignment metric (BAM) [2]. c: The pairwise distances between each pair of curves are stored in the shape similarity matrix. d: The eigenvectors for the first 5 eigenvalues of the shape similarity matrix are calculated to create the diffusion map embedding. The majority of the shape information is captured by the first two diffusion coordinates. e: Hierarchical clustering using affinity propagation with seriation extension can be used to identify similar shapes independently of diffusion map embedding. f: The hierarchical cluster membership can be colour-coded in shape space, showing how clusters are conserved within the shape space map. Exemplars for each cluster are shown. For example, for the dataset used here the red cluster contains elongated cell shapes, while the blue cluster contains more rounded shapes. g: The shapes for a single cell can be tracked through the shape space and centroid positions plotted to illustrate their migration through real space.

shapes tested. BAM calculates the minimal $L^2$-norm between both curves, i.e. minimises the sum of the squared distances between corresponding points, following translation, rotation and renumbering of nodes to achieve optimal alignment and node pairing in both curves. To retain any shape differences between left and right-turning cells, we did not include reflection as a permissible curve transformation. We find that translation of all curves to lie with their mean at the origin minimises the $L^2$-norm over all possible translations. The output of BAM is a shape similarity matrix with the BAM distances between each pair of curves (Fig 1c).

**Algorithm 1 Computing the best alignment metric between pairs of curves in a large dataset.**

**Input:** $\mathcal{U}$, a set of $M$ planar curves. Each curve is represented by a cyclic sequence, $v = v_0, v_2, \ldots, v_{N-1}$, of $N$ equally spaced complex numbers with mean equal to zero. **Output:** $D$, an $N \times N$ dissimilarity matrix.

*for-loop* over $v \in \mathcal{U}$

1. Compute $s(v) = \sum_{i=0}^{N-1} |v_i|^2$
2. Compute $\left(c_{v,j}\right)_{j=0}^{N-1}$, the fast Fourier transform of $\left(\overline{v_j}\right)_{j=0}^{N-1}$.
3. Compute $\left(f_{v,j}\right)_{j=0}^{N-1}$, the fast Fourier transform of $\left(v_{(N-j-1)}\right)_{j=0}^{N-1}$.

*end*

*for-loop* over $v \in \mathcal{U}$

 *for-loop* over $\nu \in \mathcal{U}$

 1. Compute $\left(X_j\right)_{j=0}^{N-1}$, the inverse fast Fourier transform of $\left(c_{v,j} f_{\nu,j}\right)_{j=0}^{N-1}$.
 2. Compute $A = \max_j |X_j|$.
 3. Compute $D(v, \nu) = \sqrt{s(v) + s(\nu) - 2A}$.

 *end*

*end*

**Diffusion map embedding.** The Diffusion Maps (DM) framework is a nonlinear dimensionality reduction technique that generates a low-dimensional coordinate representation of data such that similar data points in the high-dimensional space are represented by new low-dimensional points that are close to each other. To perform a DM-based low-dimensional embedding of n contours, $\{f_i\}$ where $1 \leq i \leq n$, we make use of the simple Gaussian kernel $\omega$ for curves $v$ and $\nu$,

$$\omega(v, \nu) = \exp \frac{-d^{BAM}([v], [\nu])^2}{2\sigma^2} \tag{3}$$

where $d^{BAM}$ is our proposed similarity measure and $\sigma$ is the chosen kernel bandwidth.

The selection of kernel bandwidth is a very important choice in this context and appears in many other contexts in computer science. The interpretation of this parameter is contextual scale, *i.e.* at what distance is something described as close and what is described as far away? In ShapeSpaceExplorer, $\sigma$ is set to be the median of all pairwise distances, which has been shown to be robust to outliers [12].

At the core of this algorithm is an eigendecomposition with the eigenvectors representing the coordinates in the final shape space representation. While the number of eigenvectors and hence the dimensionality of the dataset can be chosen to reflect the data, during development 5 eigenvectors were found to be sufficient (Fig 1d and S1 Fig).

**Out of sample extension.** In some experiments, it may not be suitable to calculate the shape space for all data at the same time. Expanding the dataset and recalculating the shape space will possibly result in a change of the diffusion coordinates. This might not be desirable if different treatments are compared to a large base dataset. To overcome this issue, ShapeSpaceExplorer includes an out of sample extension tool, which allows new data to be added without changing the learned geometry of shape space. To embed new data into an existing DM, $K$ Nearest Neighbours is used as a regression model by simply calculating the average of the numerical target of the $K$ nearest neighbours to a new input

sample. Choosing the optimal value for $K$ is critical and is best done by first inspecting the data. For example, a large $K$ value is more precise as it reduces the overall noise but is computationally very expensive. In this work, we have selected $K = 5$ for accurate and fast assignment. We also implemented a second approach for out-of-sample-extension using Laplacian Pyramids that performs comparably. The user can select the desired method and specify the value for K if using the K-Nearest Neighbour method.

### Data exploration

To enable users to efficiently and intuitively explore their data via the Shape Space, the software includes several tools. This includes:

- Shape feature calculation - properties of all the shapes in the Shape Space can be calculated and values plotted within the shape space.
- Shape space slicing - divide the shape space into equal slices based on their diffusion map coordinates and visualise the average shape of each row and column. In combination with group analysis, distribution of data in each subsection can be analysed.
- Interactive shape slicer - users can interactively select regions from the Shape Space and view the average shape.
- Affinity propagation clustering - affinity propagation and seriation algorithms are implemented to automatically classify similar shapes together (Fig 1e and 1f). This can be used for independent validation and combined with group analysis.
- Self organising map clustering - creates zones in the Shape Space using an unsupervised neural network. Distribution of data in each zone and trajectories of cells that move from one zone to another can be analysed with group analysis.
- Shape space dynamics and migration - the path through real space and shape space can be visualised for each cell ID in a graphical user interface (Fig 1g).

The group tool allows users to separate their input data by experimental conditions, such as control vs treatment or mutant.

In the next section, we demonstrate how the shape space representation and tools provided by ShapeSpaceExplorer can be used to quantitatively analyse cell morphology; the dynamics of cell shape changes; and how cell shape correlates to cell migration.

## Results

The outline of cells from image sequences of RPE1 cells expressing mGFP-LifeAct were extracted using the contour extraction approach described above within the ShapeSpaceExplorer graphical user interface (Fig 1a). Manual corrections were made to segmentation errors by i) excluding cell outlines that touch the borders or overlap with other cells; ii) merging fragments of long cell extensions to prevent bias in the dataset; and iii) removing those cells that undergo mitosis or apoptosis. The resulting dataset of 37,818 shapes of migrating retinal pigment epithelial (RPE) cells required $1.4 \times 10^9$ pairwise shape comparisons. The BAM distances (Fig 1b) were then translated into a shape similarity score using a simple Gaussian kernel (Fig 1c). The Diffusion Maps algorithm used the resulting shape similarity matrix for the low-dimensional representation of the cell shapes [13]. Calculating $1.4 \times 10^9$ BAM distances and performing the diffusion map embedding for five dimensions was completed within five hours on a high performance desktop computer (Windows 10, Intel Core i9-7900X CPU and 128 GB of RAM running Matlab 2022b) using the sparse implementation to find the eigenvalues and eigenvectors (this is included as an option in the software). For this dataset, we find that the first two dimensions of the shape space capture 86% of the shape variation and are sufficiently accurate to represent shape variation in our dataset of migrating RPE cells (Fig 1d).

2D closed curves are often analysed with Fourier Descriptors. We believe that BAM is better suited for shape space mapping because it is reversible and we can therefore generate a shape from an arbitrary point in shape space, while during the computation of the power spectra for Fourier Descriptors the phase information is lost and therefore, there is no unique solution for the reverse transformation. Calculating the Euclidean distance between Fourier Descriptors as shape similarity measure was about 4.5 times faster than BAM. However, using both approaches to identify the 5 most similar cell shapes in our dataset relative to each of 10 randomly selected examples shows that BAM performs better and consistently picks out shapes with characteristic features (Fig 2a).

## Visualisation and quantification of cell morphology in shape space

We performed two independent strategies to assess the performance of our shape space map: (1) we performed affinity propagation followed by seriation and hierarchical clustering to sort cell shapes independently of the Diffusion map algorithm, and (2) we visualised the continuous distribution of shape features. The 485 exemplars resulting from Affinity propagation (AP) clustering and 4 higher order clusters generated by hierarchical clustering of the AP exemplars are preserved with high integrity in our shape space representation (Figs 1e, 1f, 2b, and 2c). This also applies to a dataset of highly diverse 2D outlines of animals and inanimate objects (S1 Fig).

Calculating the average cell shapes in slices of the shape space map enables visualising the gradual change in shape along the diffusion map coordinates (Fig 3a). This shows that the first coordinate predominantly captures the elongation

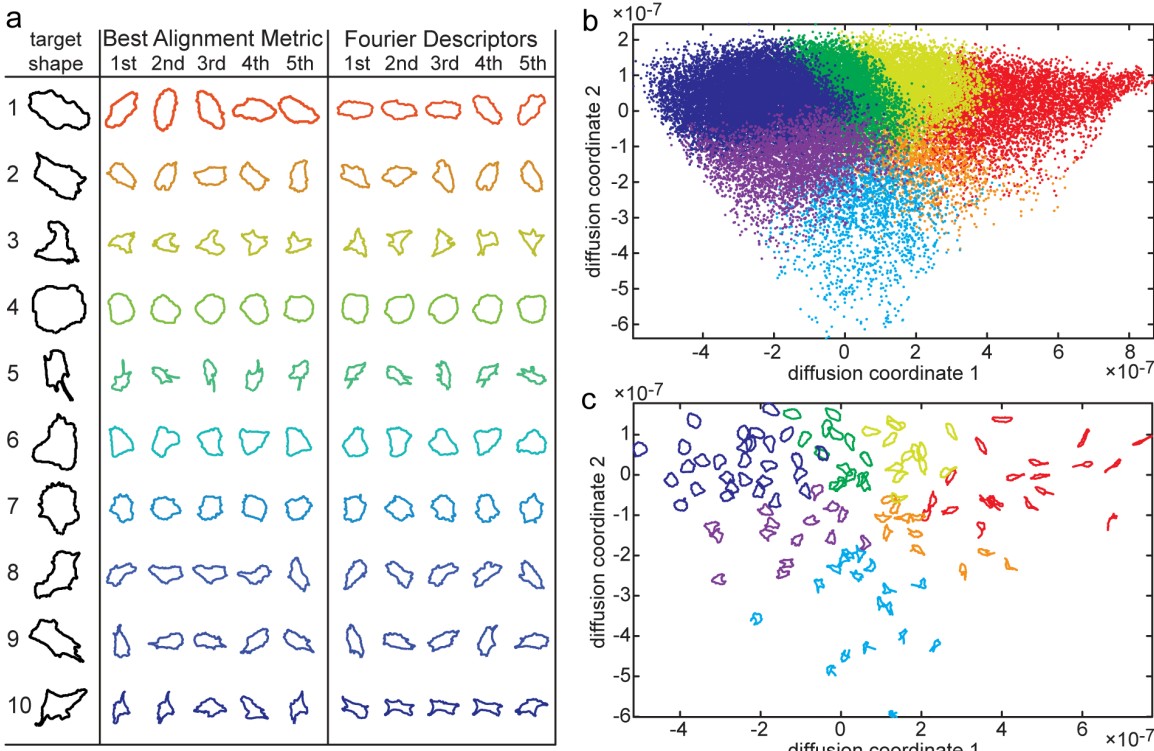

**Fig 2. Comparative performance of shape distance measures.** a: Each row shows a randomly selected target RPE cell outline, followed by 5 outlines identified as closest to the target outline (excluding outlines of the same cell as the target) according to the Best Alignment Metric and Fourier Descriptors. Note that BAM reliably selects shapes with the same typical characteristics such as central tail on D-shaped cell in line 5, gentle left bend of cell in line 8, straighter and longer left side of cell in line 2. b-c: Diffusion map embedding with cluster membership colour-coded in shape space showing all shapes as dot in b and exemplar shapes plotted at their respective position in shape space in c.

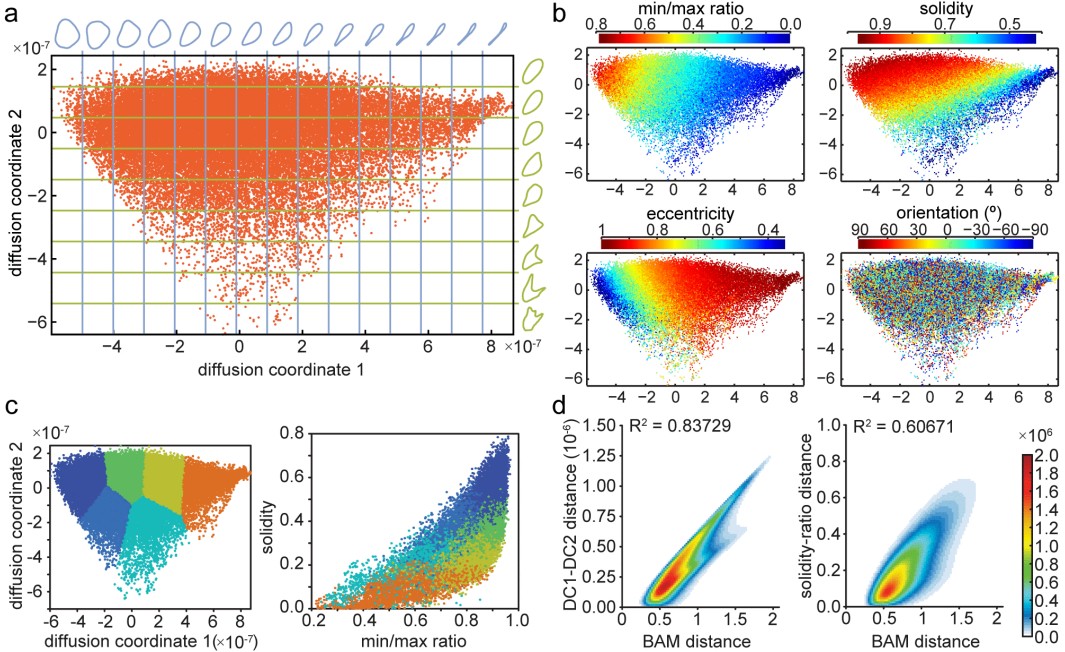

**Fig 3**. **Diffusion coordinates and BAM values correlate to shape features.** a: Average cell shapes for shape space sliced into a regular grid. This visualises which shape changes are captured by each of the diffusion map coordinates. b: Shape features colour coded for each shape in shape space: the ratio of minimal and maximal distances of a point on the curve to the centre of mass (min/max ratio), ratio of cell area to area of convex hull (solidity), ratio of minor and major axes length of an ellipse fitted to the shape (eccentricity) and the angle of the major cell axis relative to x axis in real space (orientation; used as shape-independent control). The calculation of these properties is described in the supplementary materials (S1 Text). X and Y axes show diffusion coordinates 1 and 2 multiplied by $10^7$. c: The shape space is divided into regions of similar shapes using a self organising map (see S1 Text for details). The same region colours are used when plotting solidity of shapes against the min/max ratio. d: Histogram of shapes divided into 100x100 bins displaying the relationship between the Euclidean distance of two shapes in our diffusion map with the BAM distance. In comparison, the Euclidean distance of two shapes in a solidity versus min/max ratio plot are correlated to the BAM distance. Correlation coefficient is displayed at top of each plot.

of the cells with round cells at low values of diffusion coordinate 1 and elongated cells with long tails having high values of diffusion coordinate 1. The second coordinate captures primarily irregularity with symmetric shapes at high values of diffusion coordinate 2 and cells containing multiple protrusions found towards the bottom of the map (i.e. low values of diffusion coordinate 2). Several shape features were calculated for the embedded shapes and their values were colour-coded to the corresponding points in the shape space representation. Distribution of the shape features is smooth, suggesting that changes in shape have been captured appropriately in our shape space representation (Fig 3b).

Correlation analysis was carried out by computing the correlation between each of our Diffusion coordinates with the shape features using:

$$r_{x,y} = \frac{\sum_{t=1}^{N} (x_t - \bar{x})(y_t - \bar{y})}{\sqrt{\sum_{t=1}^{N} (x_t - \bar{x})^2 \cdot \sum_{t=1}^{N} (y_t - \bar{y})^2}} \tag{4}$$

where $N$ is the number of cell shapes $N$=37818, $x_t$ represents the relevant Diffusion coordinate and $y_t$ represents the shape feature value for curve $t$ and $\bar{x}$, $\bar{y}$ represent the respective means. This confirms that while cell orientation and area distribute evenly in shape space, the ratio of minimal to maximal distance of points to the centre of mass correlates best with the first diffusion coordinate, and solidity (the ratio of area to convex hull) correlates closest to the second diffusion

coordinate (see Table 1). This is consistent with the conclusions from inspecting the average shapes along each coordinate as described above whereby increasing cell elongation is captured with increasing values of diffusion coordinate 1 and decreasing values of minimal and maximal distance ratio; and irregularities such as multiple protrusions are associated with decreasing values of diffusion coordinate 2 and decreasing values of solidity.

To test whether using two shape features instead of the diffusion map embedding would faithfully represent the shape similarity in a map, we plotted solidity against minimal to maximal distance ratio. While areas with similar shapes are roughly preserved, the space is compressed and distorted suggesting that these shape parameters are dependent (Fig 3c). To compare both approaches more quantitatively, we calculated the Euclidean norm of pairwise differences for the solidity and distance ratio and the first two diffusion coordinates. When compared to the BAM Distance, the relationship is approximately linear with a significantly higher correlation factor (0.84 versus 0.61) for diffusion coordinates compared to shape feature coordinates (Fig 3c and 3d).

When comparing cell morphology in perturbation experiments, usually specific shape features are measured, requiring prior knowledge of important parameters. Using the ShapeSpaceExplorer software, morphology classes can be defined as regions in shape space (Fig 4a, 4c, and 4d) and the relative abundance of shapes in different datasets computed (Fig 4b and 4e). To do this, the shapes from different experimental conditions are either embedded into shape space at the same time or later added using out-of-sample extension (OoSE). As proof-of-principle, we performed an RNAi experiment in which we treated RPE1 cells expressing mGFP-LifeAct with siRNA against KIF1C, a molecular motor implicated in cell adhesion protein transport [2] and a non-targeting control siRNA. From this experiment, we extracted 2836 shapes from siControl and 5009 shapes from siKif1C-treated cells and embedded these into the shape space of the large dataset of untreated RPE1 cells shown in Figs 1 and 3. We then determined the fraction of shapes in the 8 regions defined by 4 equal slices along the 1st dimension and 2 equal slices along the 2nd dimension of shape space (Fig 4a and 4b). Likewise, we analysed membership in regions whose boundaries were derived with the help of a self-organising map (Fig 4c, 4d, and 4e). In agreement with previous results showing that Kif1C is required to maintain cell tails [2], we find the majority of siKIF1C-treated cells in the regions with round or slightly elongated morphology (Region 1 in Fig 4), and significantly fewer highly elongated cells with long tails were present in the siKif1C dataset compared to control cells (Region 4 in Fig 4). Thus, our algorithm allows unbiased classification and quantification of cell shapes and determining the major morphological differences between two datasets without any prior knowledge.

**Table 1**. This table shows the correlation of simple shape features with the diffusion coordinates (D.C.) from the Diffusion Maps representation of our RPE1 dataset.

| Shape Feature | D.C.1 | D.C.2 | D.C.3 | D.C.4 | D.C.5 |
|---|---|---|---|---|---|
| Area | 0.1882 | 0.1888 | 0.0869 | 0.0141 | 0.0063 |
| Major Axis Length | 0.7949 | −0.0714 | 0.0411 | 0.0835 | −0.0039 |
| Minor Axis Length | −0.3844 | −0.5346 | −0.1235 | −0.0735 | 0.0130 |
| Eccentricity | 0.8303 | 0.2197 | 0.0740 | 0.1650 | −0.0369 |
| Orientation | −0.0035 | −0.0261 | −0.0091 | 0.0111 | 0.0052 |
| Convex Area | 0.4258 | −0.4339 | −0.0096 | 0.0349 | 0.0079 |
| Solidity | −0.6427 | 0.6315 | 0.1268 | −0.0382 | 0.0024 |
| Extent | −0.7858 | 0.4063 | 0.1378 | 0.0071 | 0.0016 |
| Perimeter | 0.6643 | −0.4763 | 0.0498 | 0.0057 | 0.0115 |
| Circularity | 0.7664 | −0.4899 | −0.0040 | 0.0181 | −0.0559 |
| Symmetry | −0.4144 | 0.6222 | 0.2296 | −0.1220 | 0.0090 |
| Max distance from centre | 0.8049 | −0.2267 | −0.0853 | −0.0860 | 0.0063 |
| Min distance from centre | −0.5287 | 0.0220 | 0.1607 | −0.2799 | 0.0338 |
| Min/Max centre distance ratio | −0.9053 | 0.2401 | 0.1629 | −0.0979 | 0.0343 |
| Irregularity | 0.8530 | −0.1703 | −0.1468 | −0.0923 | 0.0033 |
| Irregularity2 | −0.5387 | 0.3960 | 0.0363 | −0.0811 | −0.0157 |

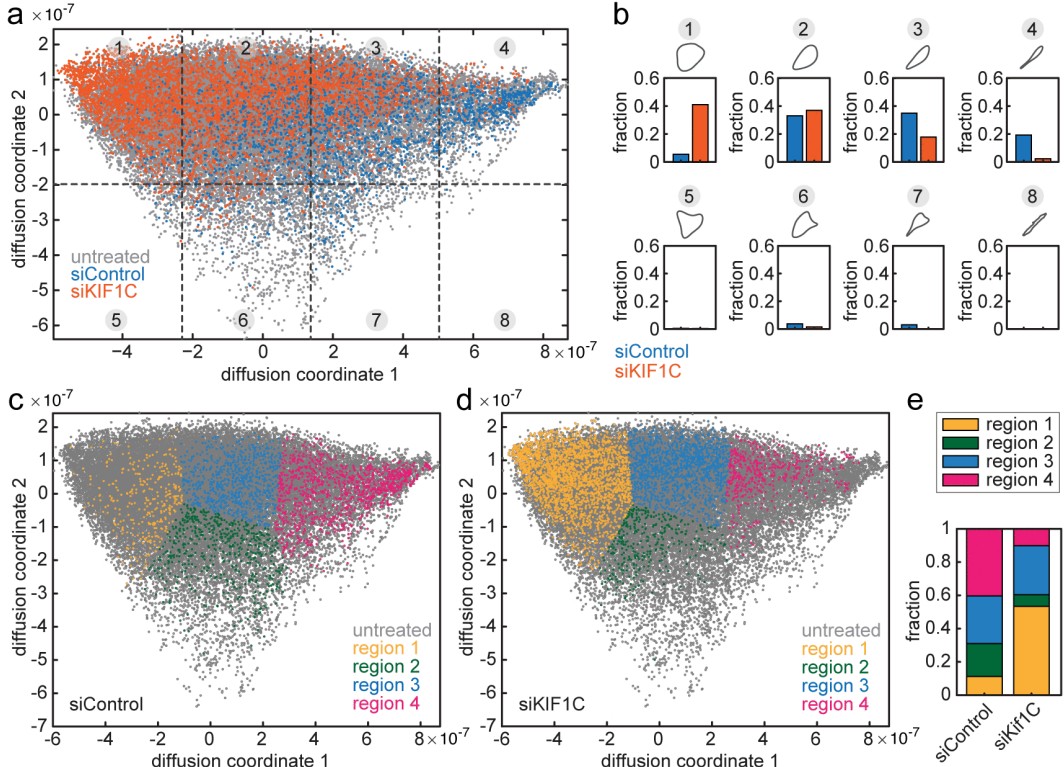

**Fig 4. Incorporation of additional data using OoSE and group analysis.** a: Shape space for the original untreated dataset (grey) with additional data from cell treated with siControl (blue) and siKif1C (orange). 8 regions obtained by equally slicing shape space with a 4-by-2 grid are indicated with dashed lines and grey circled numbers. b: Bar graphs showing the fractions of shapes, which lie in each of the 8 regions of shape space as indicated in a with the average shape for the region plotted above each graph. c-d: Diffusion map regions specified using a self organising map with data from cells treated with siControl (c) and siKIF1C (d) plotted in colour on top of the untreated dataset shown in grey. e: Stacked bar graph showing the fraction of shapes within the 4 regions indicated in c and d.

## Visualisation and modelling of shape dynamics

We considered that to understand cell migration, the distribution of shapes alone would not be sufficient, but rather the dynamics of changing from one shape to another. The 2-dimensional shape space map allows easy visualisation of the trajectory of single cells through shape space (Figs 1g and 5a). Differences in shape dynamics can be detected as the speed and directionality of trajectories through shape space. In line with the idea that the first diffusion coordinate captures the largest variability in the dataset, the majority of shape transitions occur parallel to that axis (Fig 5b). While the overall speed of shape transitions is similar in RPE1 cells treated with siRNA against KIF1C and a non-targeting control (Figs 1g and 5c), spatially resolved analysis shows that tail retractions (i.e. shape transitions of highly elongated cells towards smaller values of diffusion coordinate 1) occur about 50% more rapidly in KIF1C-depleted cells than in siControl-treated cells (Fig 5d and 5e). This is in agreement with previously published data on cell tail dynamics in KIF1C-depleted cells [2].

We next asked whether analysing the temporal shape information is sufficient to predict cell migration behaviour. Cells undergo directional change using several distinct pathways transitioning through different morphology intermediates (Fig 6a). Two of these pathways are front-led, with either a drifting front causing a gradual change in direction and a curved cell shape during the turn or with a splitting front whereby one of the forks takes over as new front. The third pathway is caused by a competing protrusion from the side or back of the cell that takes over as the front. The fourth pathway

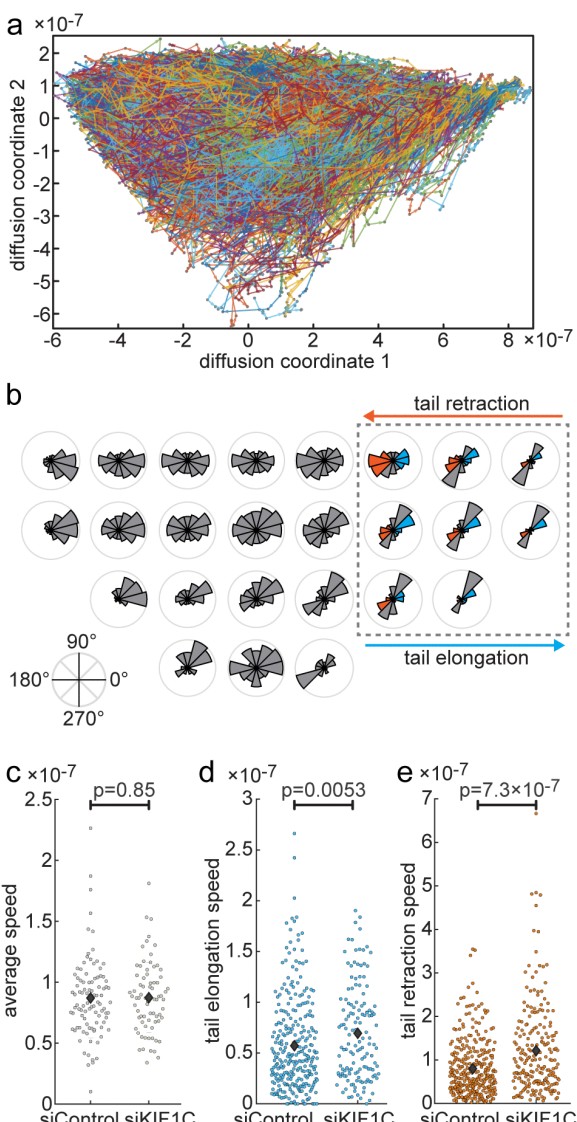

**Fig 5. Dynamics of shape changes.** a: Cells can be tracked through shape space. All the tracks are shown for the first two diffusion coordinates for the untreated RPE1 cell dataset. A subset of track segments are shown as arrows to indicate directions. b: Rose plots showing the direction of change in an 8-by-4 grid of equally sliced shape space regions for the untreated cell dataset. Regions without cell tracks are left blank. The boxed region contains cell shapes with tails (see Figs 2c and 3a) with values between −45° and 45°, i.e. towards larger values of diffusion coordinate 1 are considered shapes undergoing tail elongation, while those with values between 135° and 225°, i.e. toward smaller values of diffusion coordinate 1 are considered undertaking tail retraction. c-e: Distribution plots showing the average speed of shape changes per cell tracked in siControl and siKIF1C datasets (c), tail elongation speeds (d) and tail retraction speeds (e) as indicated with blue (elongation) and orange (retraction) in the boxed region in Fig 4b. Speed is shown as euclidean distance in shape space in a 5 min interval. n = 95, 74 cell tracks, 289, 139 elongation events, 319, 213 retraction events in siControl, KIF1C dataset respectively. Diamonds indicate mean value. p value from Kuskall-Wallis test.

requires the depolarisation of the cell, often triggered by the retraction of the cell tail and the following repolarisation in a new direction (Fig 6a). We considered that these morphological transitions are captured as trajectories in shape space and that therefore the shape space trajectories contain information on cell behaviour.

As a proof-of-principle, we set out to test whether we can detect sequences of cell contraction and elongation in our dataset to predict cell turns from the shape information alone. To do this, we trained a Hidden Markov Model (HMM) with

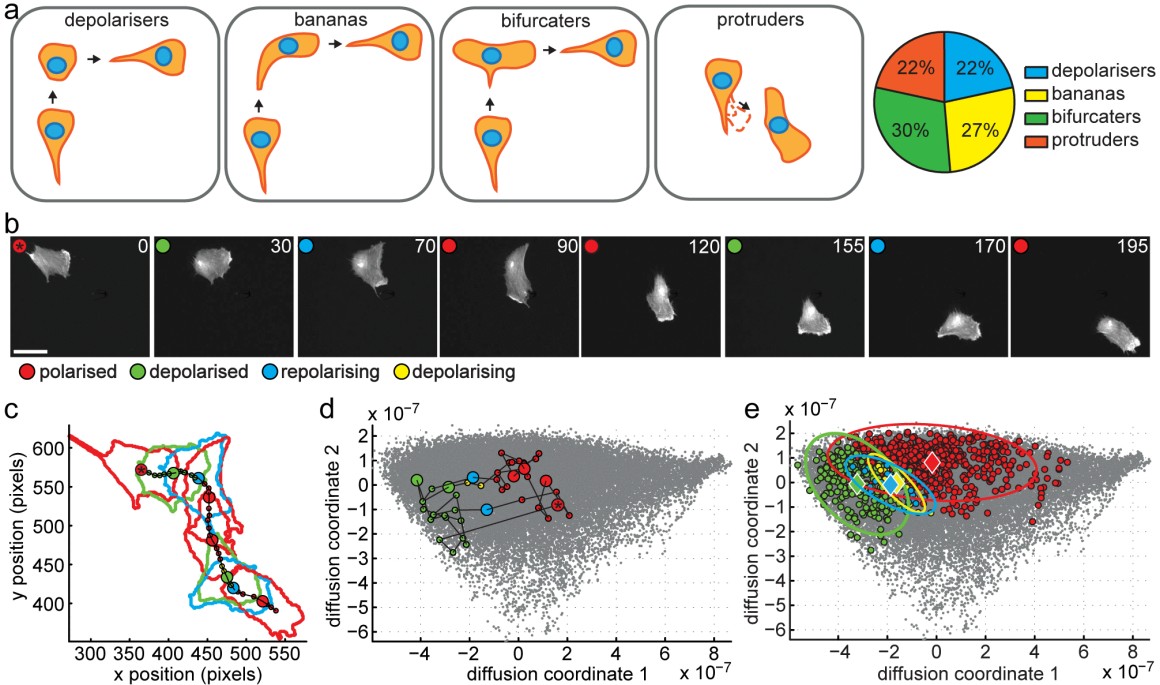

**Fig 6. Model to detect cell turns.** a: Schematic and prevalence of four different modes of cell turning in RPE1 cells: depolarisers contract and then protrude into a new direction, bananas gradually change direction of protrusion resulting in curved cell shape at turn, bifurcaters split the cell front with one front later becoming the new rear and the other the new front, protruders form a new protrusion at the side or rear that becomes the new front. All four modes account for roughly a quarter of observed cell turns in RPE1 cells expressing GFP-Actin. n=37 cells. b-d: Example of depolariser from training set for Hidden Markov Model (HMM). All 43 frames in the track were classified into one of four categories representing important stages of the mechanism: red, green, yellow and blue corresponding to polarised, depolarised, depolarising and repolarising respectively. Selected stills from timelapse movie in b, corresponding segmented outlines and the path of the 'centre of mass' of the cell in real space in c and track through the Diffusion Maps based shape representation in d. Selected frames shown in b are shown with larger dots in c and d. The asterisked red dot corresponds to the first frame in the sequence. Time is given in minutes, the scale bar corresponds to 25 µm. Grey dots in d are positions of the shapes from full RPE1 dataset described in Figs 1–3. e: Diffusion map embedding of RPE1 cells and training of 4-state Hidden Markov Model with polarised (red), depolarised (green) and two transition states depolarising (yellow) and repolarising (blue) to which a 2D Gaussian was fitted, displayed in the plots by a diamond as the mean and an ellipse representing the covariance. In our framework, we are predicting that the cell turns will coincide with repolarisation (blue) events.

10 typical image sequences showing a depolarisation-induced turn. We classified shapes into four states: polarised, depolarised, depolarising (i.e. transitioning between polarised to depolarised) and polarising (i.e. transitioning between depolarised to polarised) and fit a Gaussian to their distribution in shape space (Fig 6b, 6c, 6d, and 6e). The HMM was then used to assign the hidden states to 440 new shape sequences from our dataset of migrating RPE1 cells. We predict that directional change occurs upon repolarisation. Therefore, we made a turn prediction for the following sequence of states: depolarised - polarising - polarised with the turn point to lie at the moment of polarisation. From that prediction follows that segments between the turns should be straight. While the latter is a simplification as we would not expect to detect all 4 modes of cell turning, we continued to test how well that simple model correlates to the tracks of centroid positions in real space. We performed a local (40°) and distant (25°) angle check at the predicted turn points to detect abrupt as well as more gradual directional changes. We further determined the straightness of track segments between predicted turns (see S1 Text for details on angle and straightness checks). We found that 339 of 460 repolarisation events resulted in a turn and 500 of 889 path segments were determined to be straight segments. This suggests that 78% of the repolarisation events correspond with an angle change and examining example traces supported this (Fig 7a and 7b). Furthermore, the distribution of distant angles at the detected repolarisation events is much broader than in the middle of other

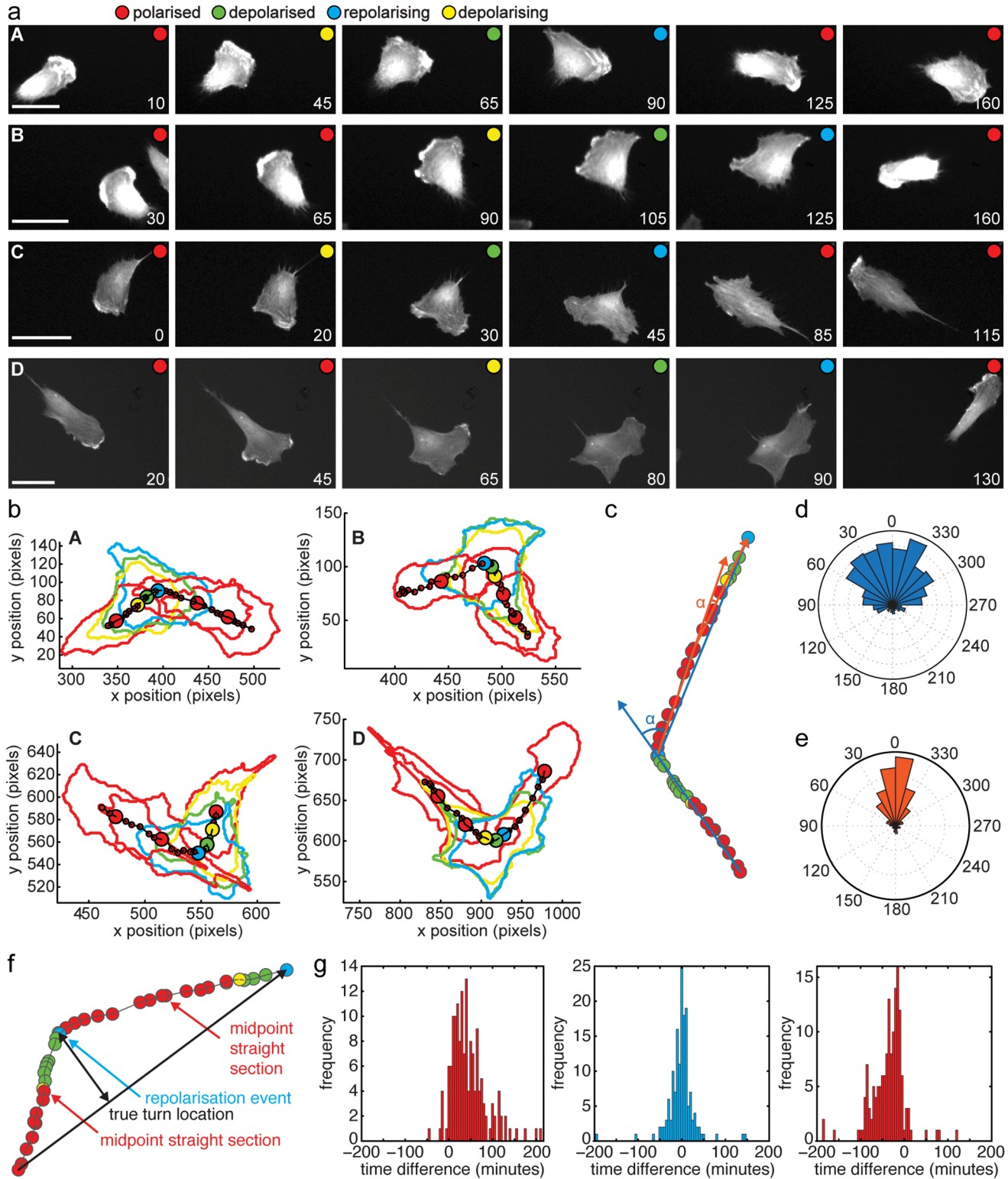

**Fig 7. Predicted cell turns.** a–b: Four cell tracks (A–D) each demonstrating a turn using the depolarisation mechanism, and each correctly identified with our HMM track analysis. For each track, six selected frames are shown. HMM classification is shown by a coloured circle as red, yellow, green and blue corresponding to polarised, depolarising, depolarised and repolarising respectively. Fluorescent micrographs of GFP-Actin is shown in a and segmented cell outlines and 'centre of mass' shown in b. Scale bar is 25 µm. Time in minutes. c: An example track to illustrate angle analysis. The path of the

centroid of a cell over time with centroid coloured according to its morphology-based HMM classification. Blue and orange lines represent the lines used for distant angle checks both at the first repolarisation event and the midpoint between two repolarisation events. d-e: Rose plots showing angles at the 460 detected repolarisation events (blue, d) and for comparison the midpoints between repolarisation events (orange, e) determined according to schematic in c. f: An example track of 'centre of mass' centroid positions to illustrate turn prediction accuracy in straight-corner-straight subsequences. The true turn location was identified as the point with the largest perpendicular distance from the direct line between first and last point of the sequence (black arrows). The time difference (5 minute intervals between data points) was determined between true turn location and predicted turn at repolarisation events (blue) and predicted turn location and the midpoint of the straight segment before the true turn and after (red). g: Time difference of predicted turn location in 155 straight-corner-straight subsequences determined as illustrated in f relative to the true turn location (middle) and the midpoint of the track before (left) and after (right) the turn.

path segments (Fig 7c and 7d). We next took 155 correctly predicted straight-turn-straight sequences and looked how precise the turning point was predicted from the repolarisation event. We defined the true location of the turn as the point that has the highest perpendicular distance to the straight line that connects the end points of the straight-turn-straight sequence. Then we measured the time delay between the predicted turn location and the true turn location (Fig 7f). For comparison, we also measured the time delay between the predicted turn location and the midpoint of the path segment before the true turn location and the midpoint of the segment after. We find that more than 61% of predicted turn points were within the 5 closest points ($\pm$10 min) to the true turn, while only 14% of the control points were close to a turn (Fig 7g). These findings demonstrate that cell shape is closely linked to cell migration behaviour. We also demonstrate that turns induced by depolarisation are a common phenomenon capturing a significant number of directional changes of migrating epithelial cells, suggesting that events at the rear of the cell have the potential to steer cell migration.

Taken together, we have proposed a framework for generating a quantitative representation of cell outlines and applied it to migrating human RPE cells. The framework is a) designed to learn the shape properties that are most responsible for the observed variation in the dataset, and so needs no prompt or supervised labelling, but can be applied to a dataset where the intrinsic degrees of freedom are not yet fully known or understood; b) has been demonstrated an effectiveness in representing a contiguous dataset where the observed variation occurs as a continuous spectrum and not as discrete clusters; and c) allows for efficient implementation to analyse large datasets. A main contribution of this work is the development of a novel shape distance measure specifically designed for the comparison of independent simple closed planar curves. This is by finding the pairwise distance between corresponding pairs of points on the curves after the curves have been mutually aligned, reparameterised, and interpolated so as to best emphasize the similarities between the curves.

This study shows that a quantitative interpretation of shape can be useful in seeking an understanding of the mechanisms of migration. Our framework has shown the ability to make predictions about the occurrence of track turns looking only at the dynamic shape information of a cell. This shows that cell shape does have a quantitatively measurable relationship with migration, and that our framework has the potential to investigate it. Beyond, cell migration, we expect that our shape explorer software package will enable the analysis of shape changes during other biological processes such as differentiation, infection and other pathological changes, and to discover molecular players that control cell morphology and the dynamics of cell morphology changes.

## Availability and future directions

The code and documentation are available in the supporting information files (S1 Data) and at https://github.com/cmcb-warwick/ShapeSpaceExplorer. The documentation pages can be directly accessed at https://cmcb-warwick.github.io/ShapeSpaceExplorer/getting_started/. The raw image data used for this study have been deposited with Zenodo under DOI: 10.5281/zenodo.14998398 and the processed data at DOI 10.5281/zenodo.14999395. Warwick's Computing and Advanced Microscopy Development Unit (CAMDU) will maintain the software and ensure that it is compatible with future versions of MATLAB. Contact details for users are provided on the github pages.

The ShapeSpaceExplorer software and the proof-of-principle that we can predict cell behaviour from cell shape changes opens up possibilities for future developments. For example, the analysis of shape space trajectories can be extended with tools to partition and classify trajectories into phases of persistent directional changes, confined diffusion (i.e. periods of minimal shape changes) and diffusive motion. Switch rates and relative time spent in these phases can be calculated for different treatments or perturbations. Further, detailed flow analysis to determine the most likely next shape from any given prior shape or sequence of shapes, might provide interesting new insights. The combined analysis of real space trajectories and shape changes would be a powerful approach to automatically classify different mechanisms of cell turning and ask how speed and directionality of cell migration correlate with morphology - potentially making it possible to predict some aspects of cell migration from still images. This should be possible with reasonable certainty in the same way as we can tell from a photograph of a person whether they were standing still, walking or running at that moment in time. Additional information, such as the total intensity or an intensity profile of a fluorescence marker within the analysed cells, could be included. Such markers could either label cytoskeletal components to further understand the relationship of cell morphology and cell migration with the underlying changes in the cytoskeleton, detect cell cycle state to understand possible changes in cell morphology and migration during G1 versus G2 phase, or any protein of interest to understand its dosage-dependent effect on cell behaviour. Additional options in the software could include reflection as permissible transformation during BAM. This would be of interest when cells turning left or right is not of interest, but outlines of objects viewed from different sides need to be robustly identified as similar. Further, an extension to interactively train Hidden Markov Models to analyse and classify user-defined shape change sequences could be included.

## Supporting information

**S1 Data. Source code.** Zip archive including complete ShapeSpaceExplorer source code and documentation. All MATLAB scripts tested with R2022b and R2024b on Windows 10.
(7Z)

**S1 Fig. BAM and diffusion map embedding of 2D shape dataset.** a-d: Diffusion map of >1000 standard shapes (https://github.com/2dshapesstructure), colour-coded by hierarchical clustering with 1st and 2nd diffusion coordinates plotted in a and b, and 3rd and 4th diffusion coordinate plotted in c and d. Exemplars plotted in their respective position in shape space in b and d. e: The Euclidean distance in shape space is calculated for 3 different shape classes (bone, cup and elephant) both to every other shape in the dataset (left) and shapes with same classification (right). Note that BAM does not include reflection and therefore, mirror images of the same shape - unless symmetrical - fall into separate groups such as left and right view of an elephant or a cups with handle curving out from top left or right.
(TIF)

**S1 Text. Supplementary methods.**
(PDF)

## Author contributions

**Conceptualization:** Samuel D. R. Jefferyes, Anne Straube.

**Data curation:** Samuel D. R. Jefferyes.

**Formal analysis:** Samuel D. R. Jefferyes, Mohammed M. Abdelsamea, Elly Straube, David B. A. Epstein, Anne Straube.

**Funding acquisition:** Anne Straube.

**Investigation:** Samuel D. R. Jefferyes, Elly Straube, Anne Straube.

**Methodology:** Mohammed M. Abdelsamea, David B. A. Epstein, Anne Straube.

**Project administration:** Anne Straube.

**Resources:** Laura Cooper, Anne Straube.

**Software:** Samuel D. R. Jefferyes, Roswitha Gostner, Laura Cooper, Mohammed M. Abdelsamea.

**Supervision:** Nasir Rajpoot, Anne Straube.

**Validation:** Roswitha Gostner, Laura Cooper, Anne Straube.

**Visualization:** Roswitha Gostner, Anne Straube.

**Writing – original draft:** Samuel D. R. Jefferyes, Laura Cooper, Anne Straube.

**Writing – review & editing:** Laura Cooper, David B. A. Epstein, Anne Straube.

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
