## [Decision Letter · Decision Letter 0]

6 Oct 2025

PCOMPBIOL-D-25-01631

ShapeSpaceExplorer: Analysis of morphological transitions in migrating cells using similarity-based shape space mapping

PLOS Computational Biology

Dear Dr. Straube,

Thank you for submitting your manuscript to PLOS Computational Biology. After careful consideration, we feel that it has merit but does not fully meet PLOS Computational Biology's publication criteria as it currently stands.  Therefore, we invite you to submit a revised version of the manuscript that addresses the points raised during the review process.  

Please submit your revised manuscript within 30 days Dec 06 2025 11:59PM. If you will need more time than this to complete your revisions, please reply to this message or contact the journal office at ploscompbiol@plos.org. Please include the following items when submitting your revised manuscript:

We look forward to receiving your revised manuscript.

Kind regards,

Jian Liu

Academic Editor

PLOS Computational Biology

Dimitrios Vavylonis

Section Editor

PLOS Computational Biology

**Journal Requirements:**

6) Your current Financial Disclosure states, "Yes ↳ Please add funding details. SJ was funded by the Warwick Systems Biology Doctoral Training Centre supported by EPSRC and BBSRC. RG was funded by a postdoctoral fellowship from the Institute of Advanced Study, University of Warwick. MMA and ES were funded by the Lister Institute of Preventive Medicine award to AS. LC is funded by the University of Warwick. AS is a Wellcome Investigator (224563/Z/21/Z). None of the funders played any role in the study design, data collection and analysis, decision to publish, or preparation of the manuscript. ↳ Please select the country of your main research funder (please select carefully as in some cases this is used in fee calculation). UNITED KINGDOM - GB".

However, your funding information on the submission form indicates an alternative arrangement for the funders. 

Please indicate by return email the full and correct funding information for your study and confirm the order in which funding contributions should appear. Please be sure to indicate whether the funders played any role in the study design, data collection and analysis, decision to publish, or preparation of the manuscript.

7) Kindly revise your competing statement in the online submission form to align with the journal's style guidelines: 'The authors declare that there are no competing interests.'

**Reviewers' comments:**

Reviewer's Responses to Questions

**Comments to the Authors:**

Reviewer #1: This work involves quantitative representation of shape distributions of migrating mesenchymal cells and analysis of cell morphology during motility. It uses the diffusion maps technique to compare pairs of cell shapes. Using this framework, the authors describe a relationship between cell shape dynamics and cell migration. The need for such a tool to extract shape features of migrating cells is well acknowledged. In that regard, the work presents a software that can be used in experiments to assess cell shape changes in response to perturbations. I think it can also be applied to static cells as well.

Having said that, it is not easy to read - it would help if the methods were described in greater detail. I am not sure if I can clearly see how this work establishes the claimed relationship between cell shape dynamics and migration. The manuscript appears written for readers already well-versed in machine learning terminology, which makes it difficult for those without such a background to follow. It took me quite some time to go through, and I still could not fully understand all aspects. Unfortunately, I do not have access to MATLAB, so I could not see the software in action, which would probably have given me a fairer impression of the work. Some specific comments are below

1. Which types of images does this work on? Does it work for all 2D images, say bright field microscopy, fluorescence microscopy, or also for 3D confocal etc.?

2. Even though the definition of the shape similarity using BAM is quite intuitive, it will be good to show how good the BAM metric is by showing it on some standardized contours. This will also help justify the use of the optimal number of nodes (512).

3. The context of the use of the Diffusion Map embedding is not clear to me. Is it to represent all the cell shape data (from a sequence of cell images) in a low dimensional form? For example, it will help if it is explicitly states what does x and y represent in equation (3) in this context.

4. Given the closed loop nature of the cell contours, why can’t Fourier series representation be used as a low-dimensional representation?

5. I also could not understand the motivation behind the calculation of pairwise alignment for all shapes (lines 115-116).

6. Shouldn’t similarity be calculated between consecutive shapes?

7. It is not clear to me what exactly is meant by the “shape space splicing” (line 129).

8. Table 1 describes the physical characteristics of the cell along with diffusion map coordinates. It will help the reader if this is discussed in little more detail. How do we relate the DC_i’s with physical characteristics?

9. Equation (4) calculates Pearson correlation. Are all the curves independent? Will the independence or dependence affect the correlation analysis?

10. I am afraid I do not find it apparent from the figure “and significantly fewer highly elongated cells with long tails were present in the siKif1C dataset compared to control cells”. (line 219)

11. In Fig. 3, are untreated cells same as control?

12. Again, I do not follow the detection of “cell turning”. Can’t that be seen directly from the trajectory itself?

In Fig. 4 (a), it will help if the direction of the flow can be highlighted.

Reviewer #2: In this manuscript, the authors present a software (MATLAB-based Graphical User Interface) for the analysis of cell shapes and migration types. The method of analysis is based on a similarity matrix that quantifies the overall proximity of uniformly distributed points representing individual cell outlines. The software utilizes an effective optimization approach to calculating the similarity matrix, diffusion map embedding, unsupervised hierarchical clustering, and even a Hidden Markov Model approach for characterization of state transitions during cell migration. The manuscript is well-structured, written with care, and substantiated with comparative analyses and applications to experimental data. This software will be of interest to both experimental and computational cell biologists. I do not have any major concerns, but I do have some suggestions to consider during the revision stage.

Minor concerns/suggestions:

1. Throughout the paper, the authors provide their choices of some parameters (usually 4 or 5) without much justification. For example, “distinct cell IDs for each continuous track segment of at least 5 frames length”, “5 eigenvectors were found to be sufficient”, “we have selected K = 5 for accurate and fast assignment”. I don’t have anything against number 5, and I understand that some choices need to be made, but at least in some cases, the authors could run analyses with 4, 5, and 6 to show that the results don’t change or that the accuracy plateaus. This could go to Supplemental Materials. Alternatively, if the software provides an easy way to adjust these parameters and interactively see results, the authors should clearly state that in the text where they make such choices.

2. In Figure 4a, the authors illustrated cell tracks in shape space coordinates. I wonder if it is possible to classify these tracks (based on the characteristics of the tracks in the real and/or shape space). The authors opted to consider in detail only cell turning, which is interesting and valuable, but some unsupervised clustering or cell trajectories could also be important for broader applications, at least as the first step.

3. On Page 11, near the end of the first paragraph, there seems to be an incomplete sentence: “We find that more than 61% of predicted turn points were within the 5 closest points (±10 min) to the true turn, while only 14% of the control points were (Fig. 6g).”

4. The last section is titled “Availability and Future Directions”, but I can’t find anything about “Future Directions”. Either the title or the text needs to be corrected.

**Have the authors made all data and (if applicable) computational code underlying the findings in their manuscript fully available?**

Reviewer #1: Yes

Reviewer #2: Yes

PLOS authors have the option to publish the peer review history of their article (what does this mean?). If published, this will include your full peer review and any attached files.

Reviewer #1: No

Reviewer #2: No

**Figure resubmission:**
---

## [Editor Report · Decision Letter 1]

22 Dec 2025

Dear Prof Dr Straube,

We are pleased to inform you that your manuscript 'ShapeSpaceExplorer: Analysis of morphological transitions in migrating cells using similarity-based shape space mapping' has been provisionally accepted for publication in PLOS Computational Biology.

Best regards,

Jian Liu

Academic Editor

PLOS Computational Biology

Dimitrios Vavylonis

Section Editor

PLOS Computational Biology

---

## [Editor Report · Acceptance letter]

PCOMPBIOL-D-25-01631R1

ShapeSpaceExplorer: Analysis of morphological transitions in migrating cells using similarity-based shape space mapping

Dear Dr Straube,

I am pleased to inform you that your manuscript has been formally accepted for publication in PLOS Computational Biology. Your manuscript is now with our production department and you will be notified of the publication date in due course.

With kind regards,

Anita Estes
